# Psychological and Cognitive Effects of Long COVID: A Narrative Review Focusing on the Assessment and Rehabilitative Approach

**DOI:** 10.3390/jcm11216554

**Published:** 2022-11-04

**Authors:** Rosaria De Luca, Mirjam Bonanno, Rocco Salvatore Calabrò

**Affiliations:** IRCCS Centro Neurolesi “Bonino-Pulejo”, Via Palermo, SS 113, C. da Casazza, 98124 Messina, Italy

**Keywords:** Long COVID syndrome, psychometric assessment, behavioral alterations, cognitive rehabilitation, conventional and advanced approaches

## Abstract

Long COVID is a clinical syndrome characterized by profound fatigue, neurocognitive difficulties, muscle pain, weakness, and depression, lasting beyond the 3–12 weeks following infection with SARS-CoV-2. Among the symptoms, neurocognitive and psychiatric sequelae, including attention and memory alterations, as well as anxiety and depression symptoms, have become major targets of current healthcare providers given the significant public health impact. In this context, assessment tools play a crucial role in the early screening of cognitive alterations due to Long COVID. Among others, the general cognitive assessment tools, such as the Montreal Cognitive assessment, and more specific ones, including the State Trait Inventory of Cognitive Fatigue and the Digit Span, may be of help in investigating the main neurocognitive alterations. Moreover, appropriate neurorehabilitative programs using specific methods and techniques (conventional and/or advanced) through a multidisciplinary team are required to treat COVID-19-related cognitive and behavioral abnormalities. In this narrative review, we sought to describe the main neurocognitive and psychiatric symptoms as well as to provide some clinical advice for the assessment and treatment of Long COVID.

## 1. Introduction

Because of the COVID-19 pandemic, caused by SARS-CoV-2, many individuals experience post-infection long-lasting symptoms, namely post-COVID syndrome or Long COVID [1,2]. Long COVID is characterized by the persistence of exhausting fatigue, neurocognitive difficulties such as mental fog, muscle pain, and weakness, as well as depression, lasting beyond the 3–12 weeks following SARS-CoV-2 infection [3]. According to Naik et al., the most common symptoms are myalgia (10.9%), fatigue (5.5%), shortness of breath (6.1%), cough (2.1%), insomnia (1.4%), mood disturbances (0.48%), and anxiety (0.6%) [4,5]. The persistence of the symptoms seems to be linked to immune dysregulation due to harmful inflammation, although the exact causes are still unknown [6,7]. Recently, a huge number of studies have reported immune abnormalities, such as an increase in the innate immune system activity, chronic fatigue/myalgic syndrome, encephalomyelitis [8], fibromyalgia [9], cognitive dysfunction [10], depression, and other mental health disorders [11,12,13]. Concerning the latter disorders, neuroinflammation may play a key role in the onset of symptoms, through either an activation of microglia or auto-immune reactions [14,15]. Indeed, Long COVID often presents with “brain fog”, which is characterized by low energy, concentration problems, disorientation, and difficulty finding the right words [16,17]. However, long-term psychological, cognitive, or adverse mental health consequences of COVID-19 have recently begun to be recognized. Hampshire et al. [18] showed that COVID-19 could have a multi-domain impact on human cognition, as assessed using psychometric subtests. In particular, people who had recovered from the infectious disease, including those no longer reporting symptoms, may exhibit significant cognitive deficits as compared to controls, when controlled for age, gender, education level, income, racial–ethnic group, pre-existing medical disorders, tiredness, depression, and anxiety. Moreover, Ocsovszky et al. [19] found a positive correlation between the level of depressive symptoms and anxiety in a Long COVID non-hospitalized cohort. Depression and anxiety have been shown to have a negative impact on symptom perception and also contribute to a higher number of symptoms in a non-hospitalized sample, suggesting a bi-directional interconnection between the clinical and psychological factors [20,21,22,23]. Therefore, multidisciplinary rehabilitation interventions are necessary to better manage individuals with Long COVID. In fact, cognitive rehabilitation, including compensatory and metacognitive strategies, which are usually administered to patients with brain injury, can be also applied to the Long COVID population [24,25].

The aim of this narrative review is to describe Long COVID’s neurological, cognitive and psychiatric sequelae and to give some clinical advice for assessment and cognitive rehabilitation of this long-lasting syndrome.

## 2. Search Methodology

The data were collected by searching on the following databases: Cochrane Library, PEDro, PubMed, and Google Scholar, using the following keywords: “long COVID symptoms” AND “long COVID cognitive manifestations” OR “long COVID neurological sequelae” AND “neuropsychiatric symptoms in long COVID” OR “depressive symptoms in long COVID” OR “anxiety symptoms in long COVID”, AND “cognitive rehabilitation in long COVID” OR “rehabilitation in long COVID”. Moreover, we also analyzed the references of the selected articles, including only English papers, in order to obtain a complete search. The articles were evaluated according to title, abstracts, and text, and selected based on their scientific validity, as per the authors’ assessment. We include systematic and narrative reviews, randomized clinical trials (RCT), and pilot studies published between July 2020 and September 2022 which deal with the Long COVID psychological and cognitive symptoms in adult patients. Two reviewers (RDL and MB) screened 616 studies, of which 276 were selected, and after removing duplicates, we finally included 54 papers that addressed the main psychometric assessment tools and the neurorehabilitative approaches.

## 3. Neurological Manifestations of Long COVID

The neurological manifestations (NMs) of Long COVID remain an outstanding issue since the pathogenic mechanisms are poorly understood despite the high prevalence of the symptoms. The most commonly reported NMs are fatigue (72%), muscle aches/myalgia (57%), and headache (53%) [26]. Orrù G. et al. [27] also included loss of smell and loss of concentration, as well as insomnia and reduced quality of life. However, it has been pointed out that anosmia and dysgeusia are more commonly related to the acute COVID-19 infection as these specific symptoms generally resolve [28]. Conversely, symptoms such as headaches, anxiety, depression, brain fog, fatigue, and insomnia are more likely to belong to the post-infection syndrome. NM onset may be due to an association between biological and psychological factors. In fact, SARS-CoV-2 could remain in brain tissue long-term, affecting neuronal loss over time, in association with systemic inflammation and cerebrovascular changes, as recently demonstrated by Desai et al. [29]. Notably, inflammation may induce neuron injury/damage through the release of the cytotoxic and chemotactic mediators, activating the surrounding microglial cells and intensifying the microglia-mediated neuroinflammation [30]. This cytotoxins release causes neuronal loss and neurodegeneration, accounting for the cross-talk between the neurons and glial cells in the neuroinflammation status [31]. To overcome these negative implications, steroids have been successfully used, especially in the acute phase and in the most severely affected patients [32]. Moreover, to counteract prostaglandin-mediated inflammation, non- steroidal anti-inflammatory drugs may be used in different stages of the disease [33]. Some authors considered the use of Palmitoylethanolamide (PEA) in the treatment of Long COVID, showing that the compound could resolve these inflammatory processes, reducing the progression of chronic inflammation and promoting positive effects on the neurological system [34]. It has been highlighted that the peripheral activation of the trigemino-vascular system, through the inflammatory cytokine storm, is strictly involved in the development of headaches [35]. Headache, from continuous mild pain to severe migraine, seems to be one of the most common and persistent COVID-19 sequelae, and it is often accompanied with trigeminal neuralgia. Neuropathic pain due to Long COVID is underestimated compared to the other symptoms, although the main clinical features of neuropathic pain in COVID-19 patients, i.e., a prickling sensation (defined as a sensation of electric shock, burns, paresthesia, and hyperalgesia), have been well described [36,37].

Many patients complained of subtle cognitive impairment and behavioral changes that may be difficult for them to describe. These symptoms are often collectively referred to as “brain fog” or “mental clouding” [38]. However, the correlation between these self-reported symptoms and objective dysfunctions remains an unclear question. Di Stadio et al. [39] investigated the possible correlation between mental clouding and olfactory dysfunction: they found that the former might interfere with the capacity of the individual to identify odors, indirectly affecting olfactory function. In addition, subjects who suffered from mental clouding, headache, or both presented a more severe olfactory dysfunction compared to those patients without neurological complaints.

## 4. Cognitive Dysfunctions, Psychiatric Symptoms, and Behavioral Alterations

Cognitive dysfunctions are becoming the most popular symptoms in the research of Long COVID. In fact, cognitive symptoms have been reported in around 70% of the subjects [40,41]. Davis et al. [42] showed a high impact of post-COVID-19 cognitive dysfunction and/or memory impairment in daily working abilities, accounting for 86% of the sample affected. Guo et al. found a similar prevalence of cognitive symptoms: 77.8% of the patients presented difficulty in concentrating, 69% brain fog, 67.5% forgetfulness, 59.5% tip-of-the-tongue word-finding problems, and 43.7% semantic disfluency (saying or typing the wrong word) [43]. In addition, it has been shown that cognitive symptoms are more likely to develop in subjects affected by fatigue, cardiopulmonary, neurological, and autoimmune symptoms. However, in current clinical practice, it is very difficult to understand and define the “type and severity of the self-reported cognitive deficits”, such as brain fog and difficulty concentrating, and consequently to more objectively measure cognitive performance. Indeed, most studies focused on the prevalence of cognitive alterations due to Long COVID, but not on the psychometric tools to measure the cognitive domains affected by the post-SARS-CoV-2 infection [44]. To overcome this issue, Alemanno et al. measured the cognitive abilities of patients in the COVID-19 post-acute phase that had experienced severe disease, using the Montreal Cognitive Assessment (MoCA). They have observed that 80% of patients reported cognitive alterations, especially in memory abilities, executive functioning, and language skills [45]. It has been observed that 33% of individuals in an intensive care unit showed a dysexecutive syndrome associated with inattention, disorientation, and reduced planning movements in response to the verbal indications [46]. Hosp et al. found a possible correlation between the cognitive dysfunctions and neurological abnormalities revealed with positron emission tomography, showing a predominant frontoparietal hypometabolism that correlated with poor MoCA scores [47], with lower scores in verbal memory and executive functions. However, these studies are limited to severely ill and old-age patients, and it is very challenging for clinicians to determine the nature of these dysfunctions and whether they are specific to COVID-19 or are more a-specific, such as a general consequence of acute respiratory distress. Indeed, some survivors of critical disease are recognized as experiencing long-term cognitive loss [48], particularly if they experience delirium [49,50]. 

In this context, it is important to know whether these deficits may also involve younger populations, as demonstrated by Almeria et al. in patients aged 24–60. This study reported that patients with neurological sequelae had lower performance in attention, memory, and executive function, suggesting an association between symptomatology and cognitive deficits [51]. 

Recent literature regarding the long-term neuropsychiatric sequelae of COVID-19 focused on self-reported symptoms through questionnaires administered either in-person or by telephone interviews [52,53,54]. Notably, Long COVID has been found to cause anxiety and depression symptoms as well as other neuropsychiatric and cognitive sequelae [55,56,57,58,59]. Indeed, the incidence of anxiety, depression, and post-traumatic stress was 42%, 31%, and 28%, respectively, in an Italian sample [60]. Considering the alarming impact of COVID-19 on mental health, Clemente et al. investigated the correlation between the psychological status of patients who had recovered from the SARS-CoV-2 infection and their inflammatory status, showing that survivors are at risk of developing psychiatric sequelae, such as anxiety, depression, and somatization symptoms, as well as sleep disorders [61,62,63]. An interesting association has also been demonstrated between high ferritin blood levels and sleep disturbances, stress, depression, and suicidal ideation [64,65]. 

To summarize, the SARS-CoV-2 epidemic was associated with psychiatric and cognitive complications, as confirmed by several researchers [66,67,68,69], and patients with preexisting psychiatric disorders reported the worsening of previous symptoms [70,71,72,73]. These findings could support the idea that those who have experienced the COVID-19 infection may be at a higher risk for neurodegeneration and dementia. In fact, COVID-19-related cardiovascular and cerebrovascular disease may also contribute to a higher long-term risk of cognitive decline and dementia in recovered individuals [74]. 

Moreover, a vast number of studies showed that obesity and diabetes have been associated with worse outcomes during the COVID-19 infection. In fact, people with obesity have an increased prevalence of diseases such as renal insufficiency, cardiovascular diseases, Type 2 diabetes mellitus, certain types of cancers, and a significant degree of endothelial dysfunction. These conditions are major risk factors for the disease severity and mortality associated with COVID-19 [75,76]. In particular, Vimercate et al. (2021) have shown that obesity is associated also with worse Long COVID symptoms such as respiratory diseases and hypertension, suggesting that being affected by overweight or obesity is associated with prolonged symptoms after resolution [77]. Today, limited evidence has shown that the clinical and socio-demographical features of the patients (such as the number of symptoms in the first week, age, and sex) before the COVID-19 infection play a key role in the prediction of Long COVID’s duration [78]. Notably, Bellou et al. have investigated the association of 91 unique prognostic factors, divided into seven different categories, including demographic and anthropometric individual characteristics, biomarkers, symptoms, clinical signs, medical history and comorbid diseases, and medications, thus facilitating the selection of candidate predictors for a prognostic model [79]. In this context, Wang et al. found that psychological distress before the COVID-19 infection, including depression, anxiety, worry, perceived stress, and loneliness, was associated with a 32–46% increased risk of Long COVID [80]. For all these reasons, urgent in-person neuropsychological and neuropsychiatric assessments on Long COVID individuals are needed. 

## 5. Psychometric Assessment and Neurorehabilitative Approach

In the rehabilitation of cognitive dysfunctions, assessment plays a crucial role, and even more so in patients with Long COVID. Few of the aforementioned studies have explored the long-term psychological, behavioral, and cognitive sequelae of this potentially disabling syndrome. In fact, only recently have researchers begun to more objectively address the specific cognitive domains affected by this disease through the use of proper clinical scales and psychometric batteries [81]. In this context, Holdsworth et al. demonstrated that systematic cognitive testing may offer a tool to ‘rule in’ objective neurocognitive insult in the wake of this prevalent disease [54]. However, administering psychometric tests overtime to the same subjects is useful to detect changes that may confirm either persistence or resolution of cognitive symptoms and deficits [82]. Specifically, some authors used the Montreal Cognitive Assessment (MoCA) to test general cognitive status in COVID-19 patients at discharge [83]. Neuropsychological and cognitive symptoms due to Long COVID were evaluated by a complete psychometric battery to assess verbal fluency, nonverbal skills, memory abilities, executive functions, and reasoning [84,85,86]. In particular, researchers noticed that both Long COVID patients and healthy participants completed the tasks, but only the “No COVID” group completed a specific cognitive task referred to as the Test of 2D mental rotation, indicating a deficit in reasoning in the Long COVID patients’ group [87]. 

O’Connor et al. have developed the COVID-19 Yorkshire Rehabilitation Scale (C19-YRS), a useful measure for examining an LCS’s sample. C19-YRS is a 22-item patient-reported outcome measure designed to evaluate the long-term impact of COVID-19 across the domains of the activities and participation of the International Classification of Functioning, Disability, and Health and to evaluate the impact of LCS rehabilitation, including clinician-completed, self-report, and digital versions [88,89].

On the other hand, post-COVID-19 psychological and psychiatric symptoms have been investigated using numerous standardized questionnaires: the Hospital Anxiety and Depression Scale (HADS) [90] and the Generalized Anxiety Disorder-7 (GAD-7) [84], for anxiety and related symptoms. In detail, Monterrosa-Blanco recommended the use of GAD-7 during the COVID-19 pandemic; it is a widely used screening tool for anxiety with a cut-off score of 10, and it has been standardized in a sample of Colombian general practitioners [91], while the Patient Health Questionnaire-9 (PHQ-9) is for depression. The PHQ-9 is a commonly used screening tool for depression with a cut-off of 10 to consider the diagnosis. The bivariate meta-analysis of 18 validation studies identified cut-off scores between 8 and 11 as optimal means for detecting major depressive disorder [92], and the Zung Self-Rating Depression Scale (ZSDS) was used to investigate the depression symptoms in the COVID-19 pandemic among Colombian university workers [93]. Moreover, considering the high frequency of sleep alterations due to COVID-19, some authors have investigated sleep quality and insomnia using ad hoc clinical scales such as the Medical Outcomes Study Sleep Scale (MOS-SS) [94] and the Pittsburgh Sleep Quality Index (PSQI) [95]. In post-acute COVID-19 syndrome, the quality of life was investigated using the EuroQol-5 Dimension (EQ-5D) [96], while fatigue symptoms were evaluated through the FACIT-Fatigue scale, a self-report questionnaire to investigate symptoms on a five-point Likert-scale, with a sum score ranging from 0 (worst fatigue) to 52 (no fatigue) [97], as well as post-traumatic stress disorder checklist for DSM 5 (PCL-5) for post-traumatic stress symptoms [98,99]. 

Most of the preexisting tools to investigate mood, behavioral, and cognitive problems in other patient populations have been adapted to individuals with Long COVID, whilst only a few scales have been specifically developed to test this syndrome (Table 1). In particular, Klok et al. proposed the post-COVID-19 Functional Status Scale (PCFS) to measure the full spectrum of functional outcomes following COVID-19 [100]. Hughes et al. have validated a novel outcome measure for patients after COVID-19: the symptom burden questionnaire for Long COVID (SBQ-LC). The SBQ-LC includes 16 symptom scales, each measuring a different symptom domain and a single scale measuring symptom interference. Each scale measures a different symptom domain (e.g., breathing and circulation) and an “interference” scale measures the impact of a person’s symptoms on everyday life. This was developed as a comprehensive measure of the symptom burden from Long COVID [101].

High-quality instruments to measure patients’ reported outcomes are required to better understand the signs, symptoms, and underlying pathophysiology of Long COVID, to develop safe and effective interventions, and to meet the day-to-day needs of this growing patient group. There is at this time a lack of information in literature about the current effectiveness of motor and psychological repair in Long COVID subjects. However, growing evidence is demonstrating the importance of a targeted rehabilitation intervention of psychological sequelae and cognitive dysfunctions, using different techniques and systems focused on various methodological approaches, in addition to an early screening. Indeed, the treatment of Long COVID is not well defined, given that, until today, no drug therapy has been shown to improve the symptomatology. 

Only a paper [102] by our group has demonstrated the potential efficacy of Palmitoylethanolamide (PEA), an endogenous lipid mediator that has an entourage effect on the endocannabinoid system mitigating the cytokine storm, in improving COVID-19-associated symptoms, as evaluated by the PCFS (a specific tool that investigates the physical and psychological symptoms following SARS Cov-2 infection). In fact, the anti-inflammatory properties of PEA are related to its ability to regulate several genes involved in the anti-inflammatory response, such as pro-inflammatory cytokines (tumor necrosis factor [TNF]-α and inter-leucine-1β), and also to counteract in the chronic inflammation status. Although some authors attempted to give indications on treating and managing this syndrome, no agreement has been reached about the best therapeutic strategy [103]. 

Compagno et al. reported the use of a multidisciplinary rehabilitation (MDR) program, including both physical training and psychological treatment, to stimulate psycho-motor parameters in patients with Long COVID. These authors suggested that the MDR program is safe and feasible in these patients and could reduce residual symptoms and promote physical and psychological recovery [104]. Fine et al. described the guidelines statement about the Long-COVID-related cognitive symptom assessment coupled with the specific cognitive disease, as well as the specific treatment recommendations [105].

Moreover, these authors reported an interesting classification of the main traditional therapeutic interventions or tasks to stimulate specific neuropsychological sub-domains, such as attention, processing speed, motor function/speed, language, memory, mental fatigue, executive function, and visuo-spatial or visuo-construction skills [105].

The main cognitive tasks and treatments, as reported in the current literature for Long COVID subjects, concerned attention process training, structured tasks for speech-language alterations, recording talks and lectures, semantic feature analysis, training in metacognitive strategies to promote self-awareness and self-monitoring, application of compensatory strategies/aids for writing and organization, use of language-mediated strategies such as self-talk or verbalization to solve problems or remember information, and cognitive behavioral therapy [106,107]. Recent evidence has shown that Long COVID patients at home will benefit from mindfulness-based cognitive therapy using different strategies that include prescribing memory-strengthening homework to address lack of concentration, inability to remember common words, and trouble recalling the events from the previous day, such as taking notes, using a planner to record information, and dividing a task into smaller increments to prevent brain fatigue, which are common strategies that can be used with pacing, endurance, and memory [108,109] (see Table 2).

These approaches are those commonly used to train patients with severe acquired brain injury (SABI) and play a role in cognitive rehabilitation following COVID-19, including the Long COVID condition.

In line with this rehabilitative prospective, Mathern et al. have emphasized the importance of a multidisciplinary team to realize some specific neurocognitive rehabilitation programs [109], such as physiatrists, neurologists, neuropsychologists, therapists of psychiatric rehabilitation, and speech therapists who can help COVID-19 patients (see Figure 1).

In fact, according to our opinion, a multidisciplinary approach to both assess and treat the sequelae of COVID-19 is essential; this also confirmed by García-Molina et al. [111]. In addition, rehabilitation programs should also focus on developing an outpatient multidisciplinary programmatic approach to improve functional outcome and facilitate recovery [112,113]. In particular, rehabilitation programs based on endurance and balance training have yielded improvements in cognition [114], as have the use of virtual reality or PC-based tasks; VR gaming was perceived as a positive and motivating rehabilitation treatment after Long COVID, with benefits regarding stress reduction and cognitive functioning [115,116,117]. In this context, a plan for discharge should also be included to follow patients in the long-term, using psychological services. In fact, tele-rehabilitation allows us to perform home-based exercises tailored to the patients’ needs, which can be implemented based on specific areas of cognitive deficit [118].

Patients could benefit from tele-therapy or tele-counseling to address emotional and mood disturbances [119], and some studies [120,121,122,123] have demonstrated improvement with on-line intervention for older adults with neurodegenerative disease and for caregivers of patients affected by severe acquired brain injury during the COVID-19 era.

Other technologies are available and approved for clinical use but have not been extensively studied in the treatment of the cytokine storm in COVID-19 patients [124]. Indeed, brain and immune response are bidirectionally correlated, acting throughout the body as potential targets for noninvasive neuromodulatory approaches [125]. In particular, an emergent and advanced approach to improve visual and cognitive impairments due to Long COVID is represented by neuromodulation, namely non-invasive brain stimulation (NIBS) using microcurrent. Considering that visual and cognitive symptoms in Long COVID may also be a neuro-vascular problem caused by a reduced blood flow (vascular dysregulation and deoxygenation) associated with neural hypometabolism, Sabel et al. hypothesized that the transorbital alternating current stimulation (tACS) might be able to reduce Long COVID symptoms [126]. However, there is no substantial evidence of treatment for this novel neuromodulatory approach to these impairments, except for that of Thams F. et al., who elaborated an innovative protocol about the combined use of transcranial direct current stimulation (tDCS) with concurrent cognitive training in post-COVID-19 infection, without publishing their results yet [127].

## 6. Discussion

Cognitive impairment in patients with Long COVID is an under-reported and challenging issue. Indeed, most patients suffer from subjective or subclinical deficits which may be under/misdiagnosed, negatively affecting patients’ quality of life. Therefore, clinicians dealing with these “fragile” patients should be aware of their cognitive and behavioral symptoms and ask for them even when they are not spontaneously reported by the patients. Indeed, we found that psychological and cognitive alterations are common concerns of individuals with Long COVID and seem to be related to different causes as well as pre-existing conditions. The activation of the inflammation and immune response with the increase in cytokines and procoagulant molecules may account for either the acute or chronic consequences of the infection [102], especially fatigue and myalgia. Neuroinflammation and the related neural loss with the reduction in microcirculation are instead believed to subtend the neuropathological basis of the Long COVID-related cognitive impairment. Conversely, most of the behavioral and psychological alterations seem to not have a biological basis but are related to the reaction to the disease. In this sense, traits of personality, coping strategies, and premorbid conditions may worsen psychiatric symptoms [15]. With these premises, it appears fundamental to assess COVID-19-induced neuropsychiatric symptoms early in order to better manage this long-lasting syndrome.

In our opinion, psychologists and psychiatric therapists should be trained on which test they have to administer in specific conditions, taking into consideration the amount of clinical and psychometric tools available. This review is based on the presentation of the main psychological and cognitive measures used in the evaluation of Long COVID syndrome, illustrating the newly designed tools (i.e., C19-YRS, PCFS, and SBQ-LC) and the “older”, adapted tools, which are commonly administered to patients with cognitive impairment due to different etiology (such as acquired brain injury and neurodegenerative disorders). In fact, although many tests, such as MMSE, MoCa, Digit Span, TMT, ZSDS, GAD- 7, etc., may be successfully used to investigate cognitive impairment as well as behavioral abnormalities in patients with Long COVID, newly specific tools are needed to better investigate these symptoms. Post-COVID-19 patients have several symptoms that may have features (duration, intensity, comorbidities, …) which are different from other pathologies and should be specifically addressed and treated.

On the other hand, we analyzed evidence about the main rehabilitative approaches, including the conventional and the advanced, that are applied to this patient population. Notably, we observed that conventional rehabilitative interventions [105,106,107,108] and strategies [112] to treat the cognitive and neuropsychiatric manifestations of COVID-19 are based on adapted programs already performed in other neurological diseases [105,114]. According to Fine et al. [105], treatment recommendations for the management of acquired brain injury-related cognitive deficits may be helpful in implementing treatment as well as the education of patients with Long COVID. Otherwise, the use of innovative technology (i.e., the PC-based approach, tele-rehabilitation, and virtual reality training) is collecting considerable interest from researchers [116,119,121]. However, the evidence on the role of these innovative approaches in improving cognitive recovery and psychological well-being in the Long COVID population is still poor. For these reasons, more consistent studies and significant data are needed, according to specific inclusion criteria such as socio-demographic variables, the duration of Long COVID syndrome, the description of symptoms, and comorbidity [78,79]. Finally, non-invasive brain stimulation (NIBS) seems to play a key role in the improvement of psychological well-being, mood symptoms [124], and sensory–cognitive deficits [126] in individuals suffering from Long COVID syndrome. In the near future, the implementation of a combined approach of tDCS associated with cognitive training [127] will be useful as a non-pharmacological and complementary approach. 

## 7. Conclusions

In conclusion, an early diagnosis of the neurocognitive and psychiatric symptoms due to Long COVID is essential in order to set up a specific rehabilitation program. The implementation in current clinical practice of early screening for the detection of post-COVID-19 could prevent enduring cognitive–behavioral alterations. The therapeutic programs must be addressed by a multidisciplinary team of experts that can give information to guide programs and public policies for the neurorehabilitation of this disabling long-term syndrome.

## Figures and Tables

**Figure 1 jcm-11-06554-f001:**
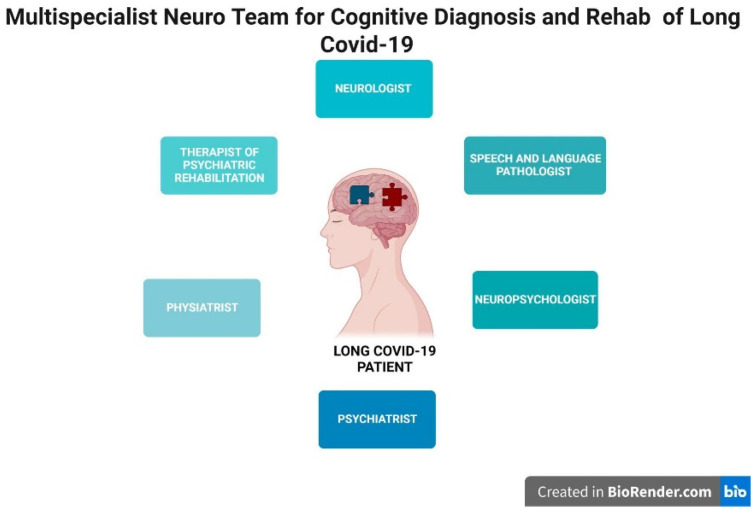
This represents the necessary healthcare figures for the cognitive diagnosis and rehabilitation of Long COVID patients. Created with BioRender.com https://biorender.com/ (accessed on 13 September 2022) [110].

**Table 1 jcm-11-06554-t001:** The neuropsychological and psychological measures used to assess cognitive and behavioral alterations in Long COVID, including the non-specific “adapted” tools and newly “designed” Long COVID ones.

Neuropsychological Measures	Domain	Short Description
**Adapted assessment tools for Long COVID**		
Mini Mental State Examination—MMSE (Fine et al., 2022) [105]	Global Cognitive Status in Long COVID subjects with moderate and severe cognitive deterioration.	MMSE is a psychometric test commonly used for screening global cognitive functioning. It consists of eleven questions and takes only 5–10 min to administer. It is a 30-point test used to measure some specific cognitive domains: − orientation to time and place (knowing where you are and the season or day of the week) − short-term memory (recall) − attention and ability to solve problems (such as spelling a simple word backwards) − language (identifying common objects by name) − comprehension and motor skills (drawing a slightly complicated shape such as two pentagons intersecting)
Montreal Cognitive Assessment—MoCA (Lynch et al., 2022) [83]	Global cognitive functioning in Long COVID subjects with mild–moderate and severe neuropsychological sequelae	The Montreal Cognitive Assessment (MoCA) is a widely used screening assessment for detecting cognitive impairment. It was validated as a highly sensitive tool for early detection of mild cognitive impairment (MCI) in 2000. MoCA accurately and quickly assesses: − Short-term memory − Visuo-spatial abilities − Executive functions − Attention, concentration, and working memory − Language − Orientation to time and place
Trail Making Test Part A and Part B—TMT A and B (Becker et al., 2021) [86]	Processing speed Executive functioning	TMT consists of 25 circles distributed over a sheet of paper. In Part A, the circles are numbered 1–25, and the patient should draw lines to connect the numbers in ascending order. In Part B, the circles include both numbers (1–13) and letters (A–L); as in Part A, the patient draws lines to connect the circles in an ascending pattern but with the added task of alternating between the numbers and letters (i.e., 1-A-2-B-3-C, etc.). The patient should be instructed to connect the circles as quickly as possible without lifting the pen or pencil from the paper. The patient is timed as he or she connects the “trail”. Results for both TMT A and B are reported as the number of seconds required to complete the task; therefore, higher scores reveal greater impairment.
Saint Louis University Mental—SLUMS (Fine et al., 2022) [105]	Cognitive impairment	SLUMS measures diverse aspects of cognition. It consists of 11 questions that help a healthcare provider evaluate: − Orientation − Short-term memory − Calculations − Naming of animals − Clock drawing test − Recognition of geometric figures − Final scores range from 0 to 30.
Mini-Cog—MC (Fine et al., 2022) [105]	Short-term memory learning	The Mini-Cog is a 3 min instrument that can increase detection of cognitive impairment in older adults. It combines a short memory test with a simple clock-drawing test to enable fast screening for short-term memory problems, learning disabilities, and other cognitive functions that are reduced in dementia patients.
Short Test of Mental Status—STMS (Fine et al., 2022) [105]	Global cognition	The Short Test of Mental Status can be administered to patients in approximately 5 min, and it contains items that test orientation, attention, immediate recall, arithmetic, abstraction, construction, information, and delayed (approximately 3 min) recall.
Digit Span—DGS (Fine et al., 2022) [105] Number span forward and backward (Becker et al., 2021) [86]	Attention and working memory	Digit Span (DGS) is a measure of verbal short-term and working memory. DGS can be used in two formats, Forward Digit Span and Reverse Digit Span. This is a verbal task, with stimuli presented auditorily, and responses spoken by the participant and scored automatically by the software. Participants are presented with a random series of digits and are asked to repeat them in either the order presented (forward span) or in reverse order (backwards span).
Digit Vigilance Test—DVT (Fine et al., 2022) [105]	Sustained attention	The DVT is a simple task designed to measure vigilance during rapid visual tracking and accurate selection of target stimuli. It focuses on alertness and vigilance, while placing minimal demands on two other components of attention: selectivity and capacity.
Cancellation test (Albert’s Test)—CT-AT (Fine et al., 2022) [105]	Unilateral spatial neglect (USN) visuo-spatial research	Albert’s Test is commonly a visual neglect screen that requires patients to cross out lines on a single piece of paper.
Letter Fluency Test—LFT adapted tool (Fine et al., 2022) [105]	Phonemic verbal fluency	Verbal fluency tests are brief assessment tools with relatively simple administration and scoring procedures. Semantic and phonemic fluency are measures of non-motor processing speed, language production, and executive functions. In particular, the Phonemic Verbal Fluency Test was shown to be sensitive for assessment of functional communication skills, commonly used for aphasic patients. Letter fluency is also referred to as phonemic test fluency.
Category Fluency Test—CFT (Guo et al., 2022) [87]	Semantic/Category fluency	Verbal fluency tests are a kind of psychological test in which participants have to produce as many words as possible from a category in a given time (usually 60 s). This category can be semantic, including objects such as animals or fruits, or phonemic, including words beginning with a specified letter, such as p, for example. The semantic fluency test is sometimes described as the category fluency test or simply as “free listing”.
Boston Naming Test—BNT (Fine et al., 2022) [105]	Denomination language abilities	The Boston Naming Test (BNT) is a widely used neuropsychological assessment tool to measure confrontational word retrieval in individuals with aphasia or other language disturbances caused by stroke, Alzheimer’s disease, or other dementing disorders. The BNT contains 60 line drawings graded in difficulty. Patients with anomia often have greater difficulties with the naming of not only difficult and low-frequency objects but also easy and high-frequency objects.
Brief Visual Memory Test—BVMT (Fine et al., 2022) [105]	Visuo-spatial learning and memory	The brief visuo-spatial memory test-revised (BVMT-R) assesses visuo-spatial learning and memory in adults. It has equivalent forms that allow reassessing of patients.
Rey–Osterrieth Complex Figure Test—R-OCFT (Fine et al., 2022) [105]	Visuo-spatial abilities, memory, attention, planning, working memory, and executive functions	The Rey–Osterrieth complex figure (ROCF) is a neuropsychological assessment in which examinees are asked to reproduce a complicated line drawing, first by copying it freehand (recognition), and then drawing from memory (recall).
Wechsler Memory Scale-IV—WMS-IV (Fine et al., 2022) [105]	Memory functions	The Wechsler Memory Scale (WMS) is a neuropsychological test designed to measure different memory functions. A person’s performance is reported as five index scores: Auditory Memory, Visual Memory, Visual Working Memory, Immediate Memory, and Delayed Memory.
State Trait Inventory of Cognitive Fatigue (STI-CF) (Fine et al., 2022) [105]	Cognitive fatigue	The STI-CF is a 32-item subjective measure of cognitive fatigue. It refers to low alertness and cognitive impairment.
Stroop Color Word—SCW (Fine et al., 2022) [105]	Executive functions	The Stroop Color and Word Test (SCWT) is a neuropsychological test extensively used to assess the ability to inhibit cognitive interference that occurs when the processing of a specific stimulus feature impedes the simultaneous processing of a second stimulus attribute, known well as the Stroop Effect.
Tower of London—TOL (Fine et al., 2022) [105]	Executive functions	The Tower of London test is a test used in applied clinical neuropsychology for the assessment of executive functioning, specifically to detect deficits in planning, which may occur due to a variety of medical and neuropsychiatric conditions.
Wisconsin Card Sorting Test—WCST (Guo et al., 2022) [87]	Executive functions	The Wisconsin Card Sorting Test (WCST) is a neuropsychological test that is frequently used to measure such higher-level cognitive processes as attention, perseverance, working memory, abstract thinking, category fluency, and set shifting. The WCST consists of two card packs with four stimulus cards and 64 response cards in each.
Word List Recognition Memory Test—WLRMT (Guo et al., 2022) [87]	Verbal memory learning	Wordlist memory tests are commonly used for cognitive assessment, particularly in Alzheimer’s disease research and screening. Commonly used tests employ a variety of inherent features, such as list length, number of learning trials, order of presentation across trials, and inclusion of semantic categories.
Pictorial Associative Memory Test—PAMT (Guo et al., 2022) [87]	Visual associative memory	Picture Memory Impairment Screen for People with Intellectual Disability (PMIS-ID). The PMIS-ID consists of four-color photographs semantically unrelated in each quadrant. It includes four distinct parts: Identification (I), Learning (L), Immediate Recall (IR), and Delayed Recall (DR)
Mental Rotation Test—MRT (Guo et al., 2022) [87]	Spatial abilities/Mental rotation	The Mental Rotations Test is a test of spatial ability. Mental rotation time is defined as the time it takes someone to find out if a stimulus matches another stimulus through mental rotation.
Hospital Anxiety and Depression Scale (HADS) (Herrmann-Lingen et al., 2011) [90]	Depression and anxiety symptoms	HADS-A consists of 7 items assessing anxiety symptoms, whereas HADS-D consists of 7 items evaluating depressive symptoms. Each item is scored on a 4-point Likert scale (0–3), providing a maximum of 21 points for each subscale. It is a patient-reported outcome measure for evaluating the emotional consequences of SARS-CoV-2 in hospitalized COVID-19 survivors with Long COVID.
Generalized Anxiety Disorder-7 (GAD-7) (Spitzer et al. 2006; Monterrosa-Blanco et al., 2021) [84,91]	Anxiety and related symptoms	GAD-7 is a screening and monitoring test for Generalized Anxiety Disorder. It is not a replacement for a diagnosis from a doctor. The answers to each question are given a value from 0 to 3 depending on severity.
Patient Health Questionnaire-9, (PHQ-9) (Olanipekun et al., 2022) [92]	Depression symptoms	The PHQ-9 is the depression module, which scores each of the nine DSM-IV criteria as “0” (not at all) to “3” (nearly every day). It has been validated for use in primary care. It is not a screening tool for depression, but it is used to monitor the severity of depression and the response to treatment.
Zung Self-Rating Depression Scale (ZSDS) (García-Garro et al., 2022) [93]	Depression symptoms	Depression was assessed using the Zung Self-Rating Depression Scale (ZSDS). This instrument consists of 20 questions split into 10 positive and 10 negative questions related to the frequency of depressive symptoms (DS). Each question receives a score between 1 and 4 (a little of the time = 1; some of the time = 2; good part of the time = 3; most of the time = 4), which means that the total score can range between 20 and 80 points, where higher scores are related to the presence of DS. For dichotomization, a global score of 55 points was taken as the cut-off point, resulting in two categories: with DS (>55) and without DS (≤55).
Medical Outcomes Study Sleep Scale (MOS-SS) (Scarpelli et al., 2021) [94]	Insomnia—Sleeping difficulty	The Medical Outcomes Study-Sleep Scale is a self-administered questionnaire with 12 items to assess sleep quality and quantity within 4 weeks (details in the online supporting information). Three variables were extracted from the MOS-SS for further analyses: (a) the Sleep Index II or sleep problem index, an aggregate measure of responses concerning four sleep domains (sleep disturbance, awakening with shortness of breath or with headache, sleep adequacy, and somnolence), as a synthetic measure of sleep quality; (b) sleep duration (item 2); and (c) self-reported evaluation of intrasleep wakefulness (item 8), dichotomized as follows: ‘‘high intrasleep wakefulness’’ (answer 3, 4, or 5) and ‘‘low intrasleep wakefulness’’ (answer 1 or 2).
Pittsburgh Sleep Quality Index (PSQI) (Taporoski et al., 2022) [95]	Sleep quality	The Pittsburgh Sleep Quality Index (PSQI) was used to assess the participants’ sleep quality. The PSQI is composed of 24 questions and measures seven different domains: (i) sleep latency, (ii) subjective sleep quality, (iii) daytime dysfunctions, (iv) sleep duration, (v) sleep disturbances, (vi) habitual sleep efficiency, and (vii) use of sleep medications, generating a global score. Each domain can be scored between 0 and 3 points, resulting in a global score ranging from 0 to 21, where higher scores are related to worse sleep quality.
EuroQol-5 Dimension (EQ-5D) (Tabacof et al., 2022) [96]	Health-related quality of life	EQ-5D is an instrument which evaluates the generic quality of life developed in Europe and is widely used. The EQ-5D descriptive system is a preference-based HRQL measure with one question for each of the five dimensions, which include mobility, self-care, usual activities, pain/discomfort, and anxiety/depression.
FACIT-Fatigue scale (Sanchez-Ramirez et al., 2021) [97]	Fatigue	Fatigue was assessed using the 13-item FACIT fatigue scale, a widely used and validated self-report questionnaire to assess symptoms on a five-point Likert-scale with a sum score ranging from 0 (worst fatigue) to 52 (no fatigue). Clinically relevant fatigue was defined by scores ≤ 30, as suggested by the creators of the scale, based on general population data.
Checklist for DSM 5 (PCL-5) (Liyanage-Don et al., 2022; Bovin et al., 2016) [98,99]	Post-traumatic stress symptoms (PTSD)	The PCL-5 is a 20-item self-report measure of the 20 DSM-5 symptoms of Post-Traumatic Stress Disorder (PTSD). Included in the scale are four domains consistent with the four criteria of PTSD in DSM-5: Re-experiencing (criterion B) Avoidance (criterion C); Negative alterations in cognition and mood (criterion D); Hyper-arousal (criterion E). The PCL-5 can be used to monitor symptom change, to screen for PTSD, or to make a provisional PTSD diagnosis.
**“Newly” Designed tools for Long COVID**		
COVID-19 Yorkshire Rehabilitation Scale (C19-YRS) (O’Connor et al., 2022; Sivan et al., 2021) [88,89]	Persistent COVID-19 symptoms	The C19-YRS outcome measure is a clinically validated screening tool recommended for use, consisting of 22 items with each item rated on an 11-point numerical rating scale from 0 (none of this symptom) to 10 (extremely severe level or impact). The C19-YRS is divided into four subscales (range of total score for each subscale): symptom severity score (0–100), functional disability score (0–50), additional symptoms (0–60), and overall health (0–10).
Post-COVID-19 Functional Status scale (PCFS) (Klok et al., 2020) [100]	Post-COVID-19 functional symptoms	PCFS is a tool to measure functional status over time after COVID-19. The PCFS scale stratification is composed of five scale grades: grade 0 (No functional limitations); grade 1 (Negligible functional limitations); grade 2 (Slight functional limitations); grade 3 (Moderate functional limitations) and grade 4 (Severe functional limitations). The final scale grade 5 is ‘death’, which is required to be able to use the scale as an outcome measure in clinical trials, but was left out for this self-administered questionnaire.
Symptom burden questionnaire for Long COVID (SBQ-LC) (Hughes et al., 2021) [101]	Long COVID burden	The SBQ-LC includes 16 symptom scales, each measuring a different symptom domain and a single-scale measuring symptom interference. It is a patient-reported outcome (PRO) measure and a multi-domain item bank that has been developed according to international best-practice and regulatory guidance. The SBQ™-LC system measures symptom burden in adults (18+ years) with post-acute sequelae of COVID-19 (PASC), also known as “post COVID-19 condition” or “Long COVID”.

Legend: MMSE (Mini Mental State Examination), MoCA (Montreal Cognitive Assessment), SLUMS (Saint Louis University Mental Status), MC (Mini-Cog), STMS (Short Test of Mental Status), DS (Digit Span), DVT (Digit Vigilance Test), CT–AT (Cancellation Test–Albert’s Test), LFT (Letter Fluency Test), CFT (Category Fluency Test), BNT (Boston Naming Test), BVMT (Brief Visual Memory Test), R-OCFT (Rey–Osterrieth Complex Figure Test), WMS -IV (Wechsler Memory Scale-IV), STI-CF (State Trait Inventory of Cognitive Fatigue), SCW (Stroop Color Word), TOL (Tower of London), WCST (Wisconsin Card Sorting Test),WLRMT (Word List Recognition Memory Test), PAMT (Pictorial Associative Memory Test), MRT (Mental Rotation Test), GAD-7 (Generalized Anxiety Disorder-7), HADS (Hospital Anxiety and Depression Scale), ZSDS (Zung Self-Rating Depression Scale), MOS-SS (Medical Outcomes Study Sleep Scale), PSQI (Pittsburgh Sleep Quality Index), EQ-5D (EuroQol-5 Dimension), FACIT (Fatigue Scale), PCL-5 (Checklist for DSM 5), SBQ-LC (Symptom burden questionnaire for Long COVID).

**Table 2 jcm-11-06554-t002:** The main conventional and advanced techniques used in the Long COVID treatment of psychological and cognitive sequelae, reporting major findings according to current evidence.

Task Domain–Specific	Conventional Approach Face to Face with Therapist Paper/Pencil Tasks		Advanced Approach PC-Based/Virtual Setting/Assistent Software Dedicated/Virtual Task	
Attention processes Well-being	Reference	Training and Major Findings	Reference	Training and Major Findings
	(Fine et al., 2022) [105]	Attention Process Training—APT (For verbal and nonverbal tasks, metacognitive strategies, timed structured activities, minimized distractions) was mentioned in the treatment recommendations, as possible therapeutic intervention strategy.	(Kolbe et al., 2021) [116]	Virtual Reality Rehabilitation The authors implemented a VR program for COVID-19 subjects. The rehabilitation unit for patients and healthcare providers was rated as highly satisfactory with perceived benefit for enhancing patient treatment and healthcare staff well-being, improving outcomes, such as mood, anxiety, sleep, pain, and feelings of isolation.
Coping, Cognition, and Mental Health	(Antonova et al., 2022) [106]	Mindfulness-based cognitive therapy These authors recommended the provision of mindfulness-based support during the COVID-19 pandemic to promote a more positive effect on well-being, as avoidance-type coping with stress and anxiety in COVID-19 context.	(O’Bryan et al., 2022) [107]	Mindfulness-based cognitive therapy via telehealth—MBCT According to these authors, MBCT (as an adjunctive treatment for anxiety via telehealth) can be considered a feasible and acceptable treatment and a promising treatment for reducing anxiety symptoms.
			(Sabel et al., 2022) [126]	Non-invasive brain stimulation using microcurrent (NIBS) NIBS can improve sensory and cognitive deficits in individuals suffering from Long COVID. Controlled trials are now needed to confirm these observations.
			(Czurra et al., 2022) [124]	Non invasive—Neuromodulation Strategies The authors have supported studies of NIBS in the current coronavirus pandemic. Neuromodulatory techniques provide a rationale for testing non-invasive neuromodulation to reduce an acute systemic inflammation and respiratory dysfunction caused by SARS-CoV-2, with beneficial role for psychological symptoms such as depression and anxiety.
Visuo-Research/Visuo-construction Skills Satisfaction Working Memory	(Fine et al., 2022) [105]	Visuo-spatial exercise programs (i.e., the use of visual cancellation tasks and strategies for visual organization, such as scanning from left to right, top to bottom, for symbols, shapes, numbers, etc.) were mentioned in the treatment recommendations, as therapeutic intervention strategies.	(Kolbe et al., 2021) [116]	Virtual Reality Rehabilitation The authors considered the use of VR that could be implemented within the context of clinical care for COVID-19 patients and both patients and staff members reported overall positive satisfaction and perceived benefit with VR as part of a comprehensive rehabilitation model.
			(Thams et al., 2022) [127]	Study protocol for a PROBE—phase IIb trial on brain stimulation-assisted cognitive training. Cognitive training group will additionally receive anodal tDCS, all other patients will receive sham tDCS (double-blinded, secondary intervention). Primary outcome: to improve working memory performance at the post-intervention assessment. Secondary outcomes: to increase health-related quality of life at post-assessment and follow-up Assessments (1 month after the end of the training).
Executive Functions Social Functioning	(Fine et al., 2022) [105]	Metacognitive strategies Problem-solving training (examples of metacognitive strategies: Goal–Plan–Do–Review, Self-talk, Goal Management Training (Stop– Think–Plan), Predict–Perform Technique). These therapeutic interventions strategies are reported as treatment recommendations by the authors.	(Maggio et al., 2020) [117]	Cognitive tele-rehabilitation home-based exercises The authors analyzed the role of cognitive tele-rehabilitation following the journalistic ‘5W’ (what, where, who, when, why), taking into account the growing interest in this matter in the ‘COVID Era’, and also promoting the practice of the human–technology interaction to improve social functioning and psychological well-being by also avoiding isolation.
			(Bernini et al., 2021) [119]	Tele-rehabilitation Home Cognitive Rehabilitation software (HomeCoRe). Authors suggested that HomeCoRe software could be incorporated as a valid support into clinical routine protocols as a complementary non-pharmacological therapy to support the continuum of care from the hospital to the patient’s home.
Communication Pragmatic Language skills	(Fine et al., 2022) [105]	Compensatory strategies/aids; writing and organization, use of language-mediated strategies such as self-talk or verbalization to solve problems or remember information (i.e., structured tasks to address various domains, such as comprehension, recall, word finding, thought organization), were described as therapeutic intervention strategies, reported in treatment recommendations.	(De Luca et al., 2020) [118]	Tele-counseling In Italy, psychological tele-counseling has been an effective method of supporting the physical and psychosocial needs of all patients, regardless of their geographical locations. In this commentary, the authors promoted the use of telehealth as an effective tool for treating patients with mental health illness, specifically, as a growing need during the COVID era.
Emotions/Mood Quality of Life Caregiver Distress	(Skilbeck et al., 2022) [108]	Cognitive behavioral therapy (CBT) This study illustrated the use of patient-led CBT for managing symptoms of Long COVID with comorbid depression and anxiety in primary care, showing reliable change in somatic, depression, and anxiety symptoms and quality of life.	(Woodall et al., 2020) [120]	Telemedicine Service/Online therapy during COVID-19. This commentary explored the use of telemedicine in reaching under-served COVID-19 patients. The authors recommended the use of telemedicine that may be helpful to limit transportation, distance, or mobility challenges, reducing physical and psychological distances.
			(Liu et al., 2020) [123]	Online therapy The authors illustrated how the main online mental health services (including online mental health education with communication programmes and online psychological counseling services) being used for the COVID-19 epidemic are facilitating the development of Chinese public emergency interventions and eventually could improve the quality and effectiveness of emergency interventions.
			(De Luca et al., 2021) [121]	Skype therapy (OLST) The authors showed that OLST may be of support in favoring global cognitive and sensory-motorrecovery in Severe Acquired Brain Injury (SABI) patients and reducing caregiver distress during COVID-19 era.
Neuropsychiatric sequelae	(Rolin et al., 2022) [112]	Rehabilitation Strategies Neuropsychiatric manifestation (psychoeducation, anxiety modulation psychotherapy, psychopharmacology, and peer support). These authors supported the idea that multidisciplinary rehabilitation of the cognitive and neuropsychiatric manifestations of COVID-19 during all levels of care is essential. An approach combining general medical, neurological, and neuropsychological intervention is recommended.	(Ghazanfarpour, et al., 2021) [122]	Tele-counseling The authors suggested that a systematic monitoring of the negative psychological impacts on medical staff is needed, as well as the implementation of appropriate tele-interventions to improve medical staff mental health of those working in hospitals and COVID-19 clinics.
Physical Function Cognition	(Daynes et al., 2021) [114]	Endurance and balance training The authors developed an adapted rehabilitation programme for individuals following COVID-19 that has demonstrated feasibility and promising improvements in clinical outcomes, with significant improvements in walking capacity, symptoms of fatigue, cognition, and respiratory symptoms.	(Groenveld et al., 2022) [115]	Virtual reality exercise at home The authors have investigated the feasibility of self-administered VR exercises at home for the post-COVID-19 condition, demonstrating that the use of VR for physical and self-administered mental exercising at home is well tolerated and appreciated in patients with a post–COVID-19 condition. Physical function outcomes registered positive health, and quality of life improved in time, whereas cognitive function seemed unaltered.

## Data Availability

Not applicable.

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
