# Peer review of "Psychological and Cognitive Effects of Long COVID: A Narrative Review Focusing on the Assessment and Rehabilitative Approach"

_jcm, 2022, doi:10.3390/jcm11216554_

Round 1

Reviewer 1 Report

- It is suggested that Table 1 is divided into “newly” developed tests for Long Covid symptoms (psychological and cognitive) and standard cognitive/psychological tests, which are generally available and not developed after Covid. If some tests which are developed during Long Covid are currently placed in the manuscript text, they can be introduced in the "new" table and divided as stated. This information is "new" in the context of this narrative review, but not solely the fact that the author repeats the list of familiar psychological tests used to assess psychological and cognitive abilities.

-The authors refer in the Introduction to psychological symptoms such as fatigue, anxiety, etc. in post covid, but in Table 1 only cognitive tests are presented (familiar to all experts in this field). Why authors did not include the available test for measuring these psychological symptoms?

-Also, it is suggested to rewrite the title of this narrative review since the paper is not focused so much on the biological and neuro side but more on the psychological and cognitive symptoms of the covid. Suggestion for the title: " Psychological and cognitive effects of Long Covid: A narrative review discussing assessment and rehabilitation approaches"

-Fig 1 the resolution can be improved and also, it is questionable if this figure brings anything new. Also, rewrite the title for Fig 1. Also, include in the title term diagnostic since these specialists do not only provide rehabilitation (therapy) but also make diagnostics. For “speech therapist,” it needs to be corrected as “speech and language pathologist,” which is equivalent to SLP as a professional diploma in this field. The symptom of Long Covid is related to language problems also (the authors refer to semantic problems as purely language problems and not speech per see), and therefore the name of those professionals needs to be corrected in figure 1. What is the difference between “therapist of psychiatric rehabilitation vs “psychiatrist”? The brain picture is it also taken from the BioRender?

-Regarding BioRender.com the author is advised to include the proper referencing in the manuscript text and Reference list.

-Fig 2, the resolution can be improved. The brain picture is it taken from the BioRender?

- Since proper validation was not conducted for all the psychological and cognitive tests on Long Covid Figure 2 is redundant in this context and it is suggested to remove this figure.  The discussion should include a summary of available recent /new psychological and cognitive tests which appeared or were developed for long covid symptoms and available approaches for therapy. Therefore part 6 should be rewritten, and instead of the title “Author’s Point of view and Conclusion,” this part should be simply called “Discussion”.

Author Response

It is suggested that Table 1 is divided into “newly” developed tests for Long Covid symptoms (psychological and cognitive) and standard cognitive/psychological tests, which are generally available and not developed after Covid. If some tests which are developed during Long Covid are currently placed in the manuscript text, they can be introduced in the "new" table and divided as stated. This information is "new" in the context of this narrative review, but not solely the fact that the author repeats the list of familiar psychological tests used to assess psychological and cognitive abilities.

We added in table one the  “newly” developed tests for Long Covid symptoms, specifying name of test/scale and describing the cognitive-behaviour domain with a short description of each assessment tool. 

-The authors refer in the Introduction to psychological symptoms such as fatigue, anxiety, etc. in post covid, but in Table 1 only cognitive tests are presented (familiar to all experts in this field). Why did the authors not include the available test for measuring these psychological symptoms? 

We now included the available test for measuring psychological symptoms (such as fatigue, anxiety etc..) in table 1. 

-Also, it is suggested to rewrite the title of this narrative review since the paper is not focused so much on the biological and neuro side but more on the psychological and cognitive symptoms of the covid. Suggestion for the title: " Psychological and cognitive effects of Long Covid: A narrative review discussing assessment and rehabilitation approaches" 

We modified the title according to the referee's suggestion. 

-Fig 1 the resolution can be improved and also, it is questionable if this figure brings anything new. Also, rewrite the title for Fig 1. Also, include in the title term diagnostic since these specialists do not only provide rehabilitation (therapy) but also make diagnostics. For “speech therapist,” it needs to be corrected as “speech and language pathologist,” which is equivalent to SLP as a professional diploma in this field. The symptom of Long Covid is related to language problems also (the authors refer to semantic problems as purely language problems and not speech per se), and therefore the name of those professionals needs to be corrected in figure 1. What is the difference between “therapist of psychiatric rehabilitation vs “psychiatrist”? The brain picture is also taken from the BioRender?

Figure 1 provides useful information about the rehabilitative process for Long Covid-19 patients, especially for those readers who are not so confident of the syndrome

We rewrote the figure title and corrected the names of those professionals. In addition, the difference between Therapist of psychiatric rehabilitation and a psychiatrist is that the first one is a healthcare worker and the second one is a physician. 

The brain picture is also taken from Biorender. 

-Regarding BioRender.com the author is advised to include the proper referencing in the manuscript text and Reference list.

We added the proper referencing in the manuscript text and Reference list referring to BioRender.com. 

-Fig 2, the resolution can be improved. The brain picture is taken from the BioRender?

Resolution was improved as requested. Yes, the template’s picture was taken from biorender, as reported in the picture footer.

- Since proper validation was not conducted for all the psychological and cognitive tests on Long Covid Figure 2 is redundant in this context and it is suggested to remove this figure.  

We removed figure 2 as suggested.

The discussion should include a summary of available recent /new psychological and cognitive tests which appeared or were developed for long covid symptoms and available approaches for therapy. Therefore part 6 should be rewritten, and instead of the title “Author’s Point of view and Conclusion,” this part should be simply called “Discussion”.

Part 6 was rewritten in terms of discussion, as suggested. 

Reviewer 2 Report

This is a narrative review of a new and interesting topic that has been in the frontline for the past two years: COVID and its long-term consequences. As such, I believe the article should be considered for publication. The authors have performed a literature review of neurocognitive effects on individuals diagnosed with SARS-CoV-2 infection. Not much has been published about it, so this issue remains controversial and a paper as this one is welcomed to shed light into the matter. However, I do have some suggestions that would help improve the paper:

Major corrections:

- Search methodology: I understand that this is a narrative review and not a systematic one. Nevertheless, I think the reader would have a better idea if the authors mentioned the number of papers that were retrieved and analyzed for this review, and the ones discarded. Ideally, these data should be summarized in a PRISMA chart, if possible, but if not, the abovementioned information would suffice.

- Certainly, inflammation plays a major role in the neurological manifestations of Long COVID; the discussion about it in subsection 3 is however somewhat superficial. I would recommend that the authors should delve more deeply into it, with particularities of inflammatory changes inside the CNS and its reversibility / possible therapeutic implications.

- Tables 1 and 2 must be restructured; in fact, all cognitive assessment tools included herein should be summarized to only facilitate understanding of COVID related results; the reader can refer to the original publication if desired. Also, such tables should include the results of the evaluations published as per each tool (table 1) and the effect of therapeutic interventions (table 2).

- If possible, the discussion regarding these same interventions should be further expanded; just mentioning that patients would benefit from them is too superficial.

Minor corrections:

The paper requires a language review; some minor grammar corrections required are:

- Abstract: muscle pain and weakness, among others

- Introduction: chronic fatigue/myalgic syndrome, controlled (not “controlling”)

- Cognitive Dysfunctions, Psychiatric symptoms and Behavioral alterations: making up for 86% of the sample… (instead of “being”), presented with, to overcome this (instead of “to this aim”), and so forth.

Author Response

In the narrative review by De Luca et al. the authors describe the main symptoms of long COVID and also they describe tools which have been used for the assessment of the syndrome. I have the following comments/suggestions:

Line 151: please correct to SARS-CoV-2 (not SARS-CoV-1).

We corrected this.

Line 210: “ regarding the potential efficacy of PEA on COVID-associated symptoms, some more discussion on the pathophysiology linking the endocannabinoid system with inflammation (and thus in this context in mitigating the cytokine storm) should be made to facilitate the unfamiliar reader.

Now we wrote further information about pathophysiology linking the endocannabinoid system with inflammation. 

Line 282: The authors should also mention the evidence regarding the use of transcranial magnetic stimulation in the context of long Covid.

Evidence regarding the use of TMS and other neuromodulation tools in the context of long covid were added. 

General comment: considering the concurrent epidemics of obesity and of type 2 diabetes and also considering the vast amount of studies showing that obesity has been associated with worse outcomes during COVID infection, the authors should comment whether obesity is associated also with worse long COVID symptoms. Also, it would be important to add in the text whether there are any prognostic factors for developing long COVID. (Is there any evidence regarding the characteristics of patients that will develop long COVID) 

We added the information about the association between obesity and worse long covid symptoms. Also we reported evidence on patients’  characteristics and prognostic factors that could increase the risk to develop long covid. 

Reviewer 3 Report

In the narrative review by De Luca et al. the authors describe the main symptoms of long COVID and also they describe tools which have been used for the assessment of the syndrome. I have the following comments/suggestions:

Line 151: please correct to SARS-CoV-2 (not SARS-CoV-1).

Line 210: “ regarding the potential efficacy of PEA on COVID-associated symptoms, some more discussion on the pathophysiology linking the endocannabinoid system with inflammation (and thus in this context in mitigating the cytokine storm) should be made to facilitate the unfamiliar reader.

Line 282: The authors should also mention the evidence regarding the use of transcranial magnetic stimulation in the context of long Covid.

General comment: considering the concurrent epidemics of obesity and of type 2 diabetes and also considering the vast amount of studies showing that obesity has been associated with worse outcomes during COVID infection, the authors should comment whether obesity is associated also with worse long COVID symptoms. Also, it would be important to add in the text whether there are any prognostic factors for developing long COVID. (Is there any evidence regarding the characteristics of patients that will develop long COVID) 

Author Response

Major corrections:

- Search methodology: I understand that this is a narrative review and not a systematic one. Nevertheless, I think the reader would have a better idea if the authors mentioned the number of papers that were retrieved and analysed for this review, and the ones discarded. Ideally, these data should be summarised in a PRISMA chart, if possible, but if not, the above mentioned information would suffice. 

As you said, this is a narrative review and not a systematic one. For this reason, we mentioned only the number of papers that were retrieved and analysed for this review, and the ones discarded. 

- Certainly, inflammation plays a major role in the neurological manifestations of Long COVID; the discussion about it in subsection 3 is however somewhat superficial. I would recommend that the authors should delve more deeply into it, with particularities of inflammatory changes inside the CNS and its reversibility / possible therapeutic implications.

We further described the role of inflammation in neurological manifestation due to Long covid in subsection 3, as suggested. 

- Tables 1 and 2 must be restructured; in fact, all cognitive assessment tools included herein should be summarised to only facilitate understanding of COVID related results; the reader can refer to the original publication if desired. Also, such tables should include the results of the evaluations published as per each tool (table 1) and the effect of therapeutic interventions (table 2).

We restructured the tables (1 and 2) as you suggested, however we reported results of the evaluations tools in the text along subsection 5, because table 1 was intended only as a descriptive one, showing to the reader a complete view of the assessment measures that could be used  in Long covid accompanied to the specific reference of the studies. Otherwise, in table 2 we reported the effects of each therapeutic intervention, as suggested. 

- If possible, the discussion regarding these same interventions should be further expanded; just mentioning that patients would benefit from them is too superficial.

We expanded discussion regarding the cognitive interventions for Long covid patients. 

Minor corrections:

The paper requires a language review; some minor grammar corrections required are:

- Abstract: muscle pain and weakness, among others; Introduction: chronic fatigue/myalgic syndrome, controlled (not “controlling”);  Cognitive Dysfunctions, Psychiatric symptoms and Behavioural alterations: making up for 86% of the sample… (instead of “being”), presented with, to overcome this (instead of “to this aim”), and so forth.

We corrected the minor grammar errors.

Round 2

Reviewer 1 Report

The suggestion is to remove the "bold" sentences in the paragraphs throughout the manuscript, there is no need to highlight the sentences since this is only a narrative review paper, and the authors are summarising knowledge on scales and some rehabilitation ways to treat long covid. Also, the suggestion is to remove the term "novelty" from the submitted paper in the discussion and rephrase the sentence. 

"The novelty of our review is based on the presentation.."

The authors used to track changes without "accepting," so it is difficult to find if fig 2 is removed.

Author Response

- The suggestion is to remove the "bold" sentences in the paragraphs throughout the manuscript, there is no need to highlight the sentences since this is only a narrative review paper, and the authors are summarising knowledge on scales and some rehabilitation ways to treat long covid.

Bold has been revoved as suggested

- Also, the suggestion is to remove the term "novelty" from the submitted paper in the discussion and rephrase the sentence. 

"The novelty of our review is based on the presentation.."

It has been rephrased

  • The authors used to track changes without "accepting," so it is difficult to find if fig 2 is removed.
  • fig has been deletaed as well as track changes.